# Mitochondrial levels determine variability in cell death by modulating apoptotic gene expression

Silvia Márquez-Jurado [1], Juan Díaz-Colunga[1], Ricardo Pires das Neves[2], Antonio Martinez-Lorente[3,4], Fernando Almazán[1], Raúl Guantes[5] & Francisco J. Iborra [1,6]

Fractional killing is the main cause of tumour resistance to chemotherapy. This phenomenon is observed even in genetically identical cancer cells in homogeneous microenvironments. To understand this variable resistance, here we investigate the individual responses to TRAIL in a clonal population of HeLa cells using live-cell microscopy and computational modelling. We show that the cellular mitochondrial content determines the apoptotic fate and modulates the time to death, cells with higher mitochondrial content are more prone to die. We find that all apoptotic protein levels are modulated by the mitochondrial content. Modelling the apoptotic network, we demonstrate that these correlations, and especially the differential control of anti- and pro-apoptotic protein pairs, confer mitochondria a powerful discriminatory capacity of apoptotic fate. We find a similar correlation between the mitochondria and apoptotic proteins in colon cancer biopsies. Our results reveal a different role of mitochondria in apoptosis as the global regulator of apoptotic protein expression.

[1] Department of Molecular and Cell Biology, Centro Nacional de Biotecnología (CNB-CSIC), Campus de Cantoblanco, 28049 Madrid, Spain. [2] UC-Biotech, Center for Neuroscience and Cell Biology (CNC), Biocant, Center of Innovation in Biotechnology, 3060-197 Cantanhede, Portugal. [3] Department of Pathology of Torrevieja and Vinalopó Hospitals, 031186 Alicante, Spain. [4] Biotechnology Department, Universidad de Alicante, 03690 San Vicente del Raspeig Alicante, Spain. [5] Department of Condensed Matter Physics, Materials Science Institute "Nicolás Cabrera" and Institute of Condensed Matter Physics (IFIMAC), Universidad Autónoma de Madrid, Campus de Cantoblanco, 28049 Madrid, Spain. [6] Program for Systems Biology of Molecular Interactions and Regulation, Institute for Integrative Systems Biology (I2SysBio), Campus Burjassot/Paterna Parc Cientific, 46980 Valencia, Spain. Silvia Márquez-Jurado and Juan Díaz-Colunga contributed equally to this work. Correspondence and requests for materials should be addressed to R.G. (email: raul.guantes@uam.es) or to F.J.I. (email: fjiborra@cnb.csic.es)

Variability in resistance of tumour cells to chemotherapeutic agents has been usually associated with genetic intra-tumoural heterogeneity. However, it is becoming increasingly clear that the non-genetic differences between cells also play a prominent role in the response and resistance of tumours to treatments[1–3]. There are many potential factors driving this non-genetic heterogeneity. Some are context dependent, influenced by the microenvironment and extracellular matrix properties surrounding the individual cells[4–6], while others are originated by differences in the internal state of each cell[7]. The relative contribution of external and internal factors is unclear and depends on the characteristics of each tumour. Nevertheless, intrinsic cell-to-cell differences are able to elicit highly variable responses by themselves. For instance, minimising context dependence by growing genetically identical HeLa cells in a homogeneous medium still shows very heterogeneous responses to drug perturbations[8] or apoptosis-inducing ligands[9]. Therefore, it is important to identify which factors are responsible for the drastic differences in phenotypic outcome when genetically identical cells are subjected to the same stimulus.

Anti-cancer apoptotic therapy eventually results in the activation of two major mechanisms, the intrinsic and extrinsic pathways, which culminate in the activation of effector caspases (Caspase-3 and 7), chromatin condensation, DNA fragmentation and finally cell death. The intrinsic pathway is directly activated by non-receptor-mediated signals, such as those caused by viral infection, toxins, free radicals or radiation. These stimuli induce mitochondrial outer membrane permeabilisation (MOMP) and the release of pro-apoptotic proteins from the mitochondria to the cytoplasm. The extrinsic route is triggered by the binding of specific ligands (FAS ligand (FASL), tumour necrosis factor (TNF) or TNF-related apoptosis-inducing ligand (TRAIL)) to the death receptors located at the plasma membrane. This binding activates Caspase-8 that directly cleaves and activates the effector caspases, and also cleaves Bid protein inducing MOMP (Fig. 1a). Therefore, there is a crosstalk between both pathways in which the mitochondria play a central role in effector caspase activation[10].

Although MOMP is considered the point-of-no-return to cell death, that rapidly releases pro-apoptotic proteins to the cytoplasm and activates Caspase-3 and 9 within a few minutes[11–13], individual cells show large variability in the time elapsed between the apoptotic stimulus and MOMP (spanning a range of 4–20 h depending on stimulus type and strength)[9,14,15]. This cell-to-cell variability in the time to apoptotic commitment has been attributed to pre-existing variability in the amount of the proteins involved in the apoptotic pathway[9].

Variability in protein levels in genetically identical cells can be originated by two different mechanisms, stochasticity in the biochemical reactions involved in the gene expression cycle (intrinsic or gene specific noise) or from fluctuations in cellular components and metabolites affecting many genes (extrinsic or

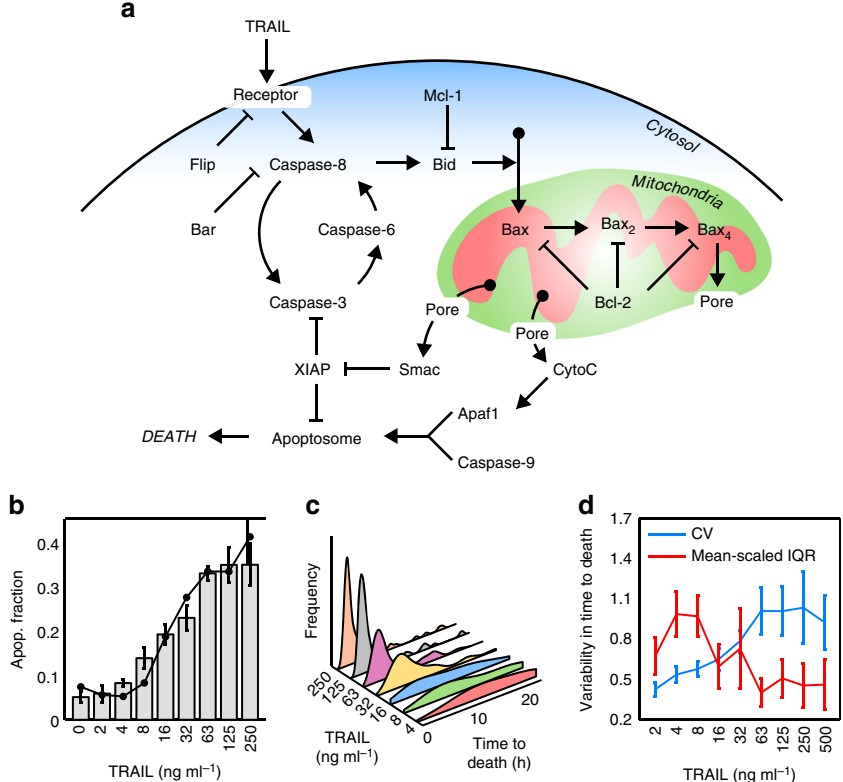

**Fig. 1** Apoptotic variability in fate and time to death of HeLa cells exposed to TRAIL. **a** Cartoon of the main protein network of the extrinsic apoptotic pathway. CytoC cytochrome C; Pore, mitochondrial membrane permeabilisation (MOMP); Bax$_{2,4}$, activation and oligomerisation process of Bax to form the mitochondrial pore. **b** Apoptotic fraction of HeLa cells after 24 h of TRAIL treatment (0, 2, 4, 8, 16, 32, 63, 125, 250 ng ml$^{-1}$). Apoptotic cells were quantified by visual inspection of phase contrast images (grey bars) and by FACS using Annexin V (FITC)-PI double staining (black dots). Around 300 cells for each TRAIL dose were inspected to obtain the apoptotic fraction. Error bars are standard deviation of three independent experiments. Data are representative of three independent experiments **c** Distributions of times to death after TRAIL treatment. Times to death were obtained by tracking cells in 24-h time-lapse experiments. Between 100 and 200 cells were analysed at each TRAIL dose to obtain the distributions. **d** Analysis of the variability in time to death at different TRAIL doses using two different statistical measures: the coefficient of variation (CV, blue) and the mean-scaled interquartile range (IQR, red). Error bars are computed by bootstrapping

global noise)[16]. The correlation in times to death observed between recently born sister cells[9, 14] suggests that the variability in TRAIL-induced apoptosis must be caused by cellular factors that globally affect gene expression[7]. Previous data from our lab showed that the heterogeneity in mitochondrial content accounts on average for 50% of the variability observed in cellular protein levels[17]. In addition, the cascade of molecular events driving programmed cell death is an energy-dependent process[18]. Mitochondrial content is highly variable from cell-to-cell[19, 20] and follows an asymmetric redistribution between daughter cells after division[20]. Recently, it has been shown that, at least in yeast, mitochondria are partitioned between daughter cells to achieve similar concentrations[21], segregating in proportion to the available cytoplasmic volume like passively segregating RNAs and proteins. Taken together these observations, the amount and/or functionality of mitochondria in individual cells could be one important cellular factor responsible for cell-to-cell differences in apoptosis times and resistance to death.

In this work, we study the impact of heterogeneity in mitochondrial content on the outcome of TRAIL-induced apoptosis in HeLa cells. We demonstrate that the amount of mitochondria is a good cellular biomarker for TRAIL sensitivity. The mitochondrial content of each cell influences the abundance of apoptotic proteins, determining its apoptotic fate and modulating its time to death. In addition, a strong correlation between mitochondrial content and apoptotic proteins levels was also observed in colon cancer biopsies, suggesting that mitochondrial mass is a good prognosis biomarker.

## Results

### Variability in the response to TRAIL-induced apoptosis.
TRAIL is a TNF family ligand that binds death receptor-4 (DR4) and DR5 on the cell surface and activates the extrinsic pathway of programmed cell death. TRAIL has been considered a promising chemotherapeutic agent due to its selectivity against tumour cells. However, many tumours show a high rate of resistance to TRAIL, severely limiting its efficiency in therapy[22]. To study the origin of this variable response to TRAIL treatment, we first assessed the apoptotic response of a clonal population of HeLa cells to variable TRAIL doses. The fraction of dead cells after 24 h of TRAIL addition was measured using two different methods, by visual inspection of phase contrast images and by FACS using Annexin V (FITC)-PI double staining. Both procedures gave very similar response curves with a sensitive region between 4 and 63 ng ml$^{-1}$ of TRAIL (Fig. 1b). For doses larger than 63 ng ml$^{-1}$, the fraction of dead cells remains approximately constant (~35%), leaving a large fraction of cells surviving to TRAIL treatment. To exclude the possibility that the fractional killing was due to TRAIL degradation or inactivation, the supernatants of different TRAIL treatments were collected and tested for apoptotic activity, showing no decreased killing efficiency (Supplementary Fig. 1). Another important observation was that many cells, both survivors and apoptotic, divided after TRAIL addition. To discard a possible effect of TRAIL on the cell cycle, we compared the distribution of division times in cells both treated and non-treated with TRAIL (Supplementary Fig. 2) and no significant effect of TRAIL on cell cycle was observed ($P > 0.3$ for two-sample Kolmogorov–Smirnov tests between control and treated samples).

Next, we focused on the variability in times to death for different TRAIL concentrations, using time-lapse movies (Fig. 1c). At low TRAIL concentrations, a large spread with similar probabilities for long and short times to death was observed, while at higher doses, average and spread of the distribution decreased, although some cells still displayed long times to death. To quantify the variability in time to death, we used two statistical

measures: the standard deviation divided by the average, or coefficient of variation (CV), and the inter-quartile range (IQR) divided by the average, which removes the effect of outliers in the spread of the distribution. Both measures showed that the variability in time to death changed with TRAIL dose (Fig. 1d). The IQR was larger at low doses, due to the effect of 'flatness' in the probability of time to death, while the CV increased at large doses, indicating a noticeable effect of a few outliers with large apoptosis times.

As TRAIL-treated cells seem to progress normally through the cell cycle, we further investigated a possible influence of cell division on apoptosis and times to death. While practically all survivor cells divided in the 24-h time-lapse after TRAIL addition (between 85% and 100% in the range of doses analysed), only a fraction of the apoptotic cells underwent division. Cells that divided before dying had longer times to death than non-dividing apoptotic cells (Supplementary Fig. 3a). Moreover, for the subset of apoptotic cells that divided, death times were positively correlated to division times (Supplementary Fig. 3b). These biases may reflect an influence of cell division on the time to apoptosis, for instance delaying apoptosis, or be simply a consequence of the fact that cells with fast commitment to death after TRAIL addition do not have time to divide before dying. To distinguish between these two possibilities, we simulated a "null" cell ensemble implying no causal relation between cell cycle stage and time to death, which reproduced both the difference in apoptosis times between dividing and non-dividing cells (Supplementary Fig. 3a) and the observed correlation between death and division times (Supplementary Fig. 3c). This indicates that apoptotic and cell cycle programs are not coupled in our system. Moreover, and in agreement with previous reports[9, 15], we found that the majority of sister cells had the same fate and very similar times to death (Pearson correlation >0.8, Supplementary Fig. 3d).

Taken together, our results confirm the presence of a threshold that must be overcome to induce MOMP[23]. The height and rate of approach to this threshold depend both on the levels of active receptors, since survival probability and death times are larger at low TRAIL doses, and on the cell's internal state, since for a given TRAIL dose both cell fate and time to death are variable from cell to cell.

### Mitochondrial content discriminates cell fate.
Mitochondria affect gene expression in a global manner[17] and are central nodes in the apoptotic route. This drove us to study the influence of mitochondrial content on the probability of cell death. We stained HeLa cells with MitoTracker Green FM (MG), which was previously shown to be a faithful reporter of mitochondrial mass[17] and has negligible phototoxic effects (Supplementary Fig. 4). Then, we treated the cells with different doses of TRAIL and imaged for 24 h at 15 min intervals. For each cell, we measured the integrated intensity of MG signal in the initial image and then we manually tracked it to assess its fate. The mitochondrial contents of surviving and dead cells were clearly different at all TRAIL doses, the cells with more mitochondria were more prone to die (Fig. 2a). This indicated that the mitochondrial level alone can be a good marker of apoptotic cell fate. To quantify the performance of mitochondrial content as a classifier of cell fate, we calculated the Receiver Operator Characteristic (ROC) curves using the probability distributions of mitochondrial mass in apoptotic and survivor cells (Fig. 2b). The area under the ROC curve (AUC) summarises the trade-off between the probability of correct and incorrect classification, and varies between 0.5 (random guessing) and 1 (perfect classifier). The ROC curves indicated that mitochondrial content is a good classifier of cell fate at all TRAIL doses analysed (Fig. 2b).

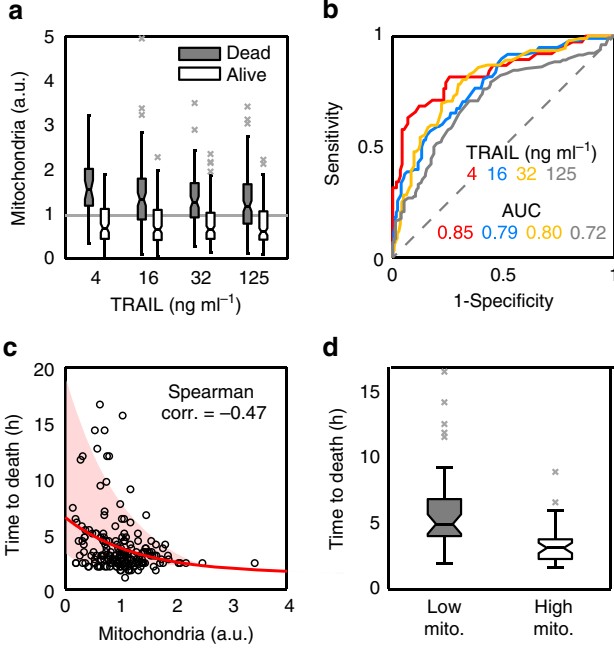

**Fig. 2** Influence of mitochondrial content on apoptotic cell fate and times to death. HeLa cells stained with MG (as mitochondrial mass marker) were treated with different doses of TRAIL. After TRAIL addition, the cells were imaged for 24 h every 15 min. For each dose, we randomly selected cells from different images, quantified their initial mitochondrial mass by integrating MG intensity, and manually tracked their fate. Typically, we gathered ensembles of 250–300 cells so as to achieve between 100 and 150 apoptotic cells per dose. **a** Boxplots of mitochondrial levels of alive (white) and dead (grey) HeLa cells after 24 h of treatment. Mitochondrial values are normalised to average (grey line). Data are representative of six independent experiments. **b** Analysis of mitochondrial content as a binary classifier (death/life) of cell fate. To calculate the performance of mitochondria as classifier, the Receiver Operator Characteristic curve (ROC) and area under the curve (AUC) were represented and calculated for the different TRAIL doses. **c** Correlation between mitochondrial levels and time to death in apoptotic single cells treated with 32 ng ml$^{-1}$ of TRAIL. The red line is an exponential fit and the shaded area indicates the confidence region for the fit. **d** Boxplots of time to death for HeLa cells with mitochondrial levels in the first quartile (Low mito, grey) and in the fourth quartile (High mito, white). Each boxplot was calculated with 30 cells. Boxes cover the range from the lower to the upper quartile of the data. Whiskers indicate maximum and minimum values, excluding outliers which are plotted as individual grey crosses. Horizontal lines inside the boxes represent median values, and notches indicate 95% confidence intervals for the median

Mitochondrial content also discriminates cell death by other apoptotic inducers like TNF-α, which triggers the extrinsic pathway, and cycloheximide (CHX) and 5,6-dichloro-1-β-D-ribofuranosylbenzimidazole (DRB) which block translation and transcription respectively, causing cell damage and activating the intrinsic apoptotic route (Supplementary Fig. 5).

**Mitochondrial mass modulates variability in time to death.** We also analysed whether there is an influence of mitochondrial mass on times to death. This effect is difficult to see since, on one hand, the mitochondrial levels of apoptotic cells are already biased to high values (Fig. 2a). On the other hand, times to death have not much variability at large TRAIL doses (Fig. 1c) while at low doses differences in receptor levels and activity may also contribute significantly to variability in apoptosis times. To avoid these

problems, we focused on an intermediate TRAIL dose (32 ng ml$^{-1}$) in the sensitive region of the dose–response curve (Fig. 1b). As shown in Fig. 2c, there is a weak but noticeable correlation between mitochondrial content and times to death (Spearman correlation −0.47). Apoptotic cells with mitochondrial levels in the first (Low) and fourth (High) quartiles showed statistically significant differences in their times to death (Fig. 2d, Wilcoxon test $p = 10^{-6}$). The cells with high mitochondrial content died on average in 3 h (with an IQR between 2 and 5 h), while the cells with low mitochondrial levels had a wider range of times to death with an average of 6 h.

In summary, these results show that apoptotic cells have significantly higher mitochondrial content than resistant cells, indicating that the amount of mitochondrial mass is a proxy for commitment to apoptosis.

**Mitochondrial content impacts apoptotic proteins expression.** Since heterogeneity in mitochondrial content is responsible for around 50% of total protein variability[17], we quantified the influence of mitochondrial mass on the amounts of transcripts and proteins involved in the extrinsic apoptotic pathway. To assess the impact of mitochondrial levels on transcripts, HeLa cells were sorted in two fractions with high and low levels of mitochondria, and total RNA was deep sequenced. As previously described[17, 24], we observed a global scaling of the transcriptome abundance between both subpopulations, where cells in the fraction with high mitochondrial levels contained around three times more RNA than cells in the low fraction (Supplementary Fig. 6). The apoptotic genes followed this general trend with fold-changes in expression around the average value of the whole transcriptome (Fig. 3a). Since changes in transcription have a variable impact at the protein level[25], we used immunolabelling to quantify the correlation between mitochondrial and protein amounts in single HeLa cells. We stained non-treated HeLa cells with a reporter of mitochondrial mass (CMXRos)[17] and different apoptotic protein antibodies. Some proteins of the apoptotic route were strongly correlated with mitochondria while others had weaker correlations (Fig. 3b). We calculated the mitochondrial contribution to variability (MCV) as a ratio of two variabilities, the variability of the original protein distribution (Fig. 3b, right panels, black distributions) and the variability of the protein distribution once the linear correlation with mitochondrial content has been removed (Fig. 3b, right panels, blue distributions). Specifically, $MCV = (1 - (CV_{det}/CV)) \times 100$, where CV is the coefficient of variation of the original distribution and $CV_{det}$ the coefficient of variation of the de-trended distribution. Similarly to other protein families[17], mitochondrial content contributed with around 50% to the total variability in the levels of apoptotic proteins (Fig. 3c). However, strong differences were detected in mitochondrial–protein correlations between the two partners of some pairs of pro- and anti-apoptotic proteins, especially the pairs Bax/Bcl-2 and Bid/Mcl-1. These data indicate that a large part of the variability observed at the protein level in the apoptotic route is a consequence of cell-to-cell heterogeneity in mitochondrial content.

**Computational model of the apoptotic pathway.** Variability in apoptotic time and fate arises from the complex interplay between pro- and anti-apoptotic proteins. To investigate how the influence of mitochondria on protein levels mediates apoptotic variability, we extended and modified a previous kinetic model of the apoptotic route that has been tested in HeLa cells[12, 26–29]. The model input is the concentration of TRAIL ligand, as well as the initial levels of key proteins of the apoptotic route (Fig. 1a). Natural variability in apoptosis probability and time to death

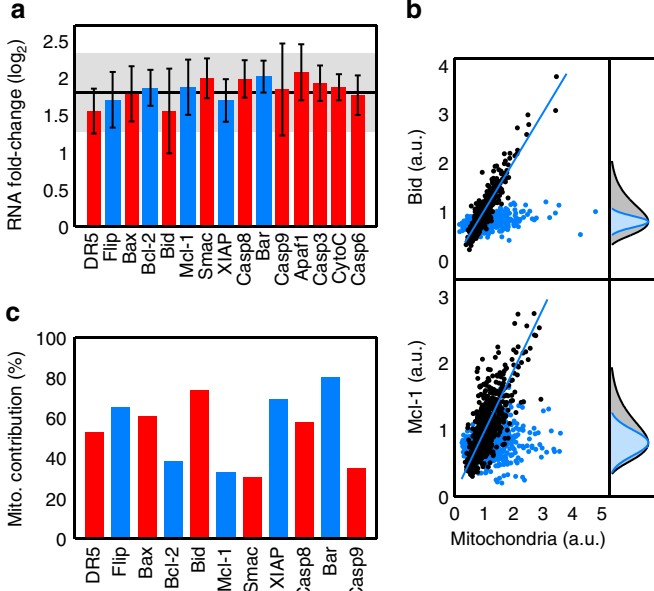

**Fig. 3** Influence of mitochondrial content on apoptotic mRNAs and protein abundances. **a** Logarithmic fold-change in mRNA expression of apoptotic genes between subpopulations of cells with low and high mitochondrial content. HeLa cells labelled with MG were sorted in two populations according to their mitochondrial content, and RNA extracted and sequenced (three independent sorting experiments were performed). The solid line corresponds to the average fold-change of the whole genome (~10,000 genes). The shaded region is the fold-change standard deviation across the whole genome. Red bars denote pro-apoptotic mRNAs and blue bars, anti-apoptotic. Error bars are standard deviations of three biological replicates for each gene. **b** Scatter plots of mitochondrial mass and protein levels in single HeLa cells (black dots). Here, we show a pro-apoptotic protein (Bid, upper panel) with a high correlation with the mitochondrial content and an anti-apoptotic protein (Mcl-1, lower panel) with smaller correlation. Blue lines represent regression lines. The corresponding distributions of protein levels are shown in grey in the right panels. The co-variation with mitochondrial content is removed (blue dots) to estimate the fraction of protein variance not due to mitochondrial levels (blue distributions). **c** Mitochondrial contribution to global variability in protein levels from different apoptotic genes, as defined in the main text. Pairs of antagonistic pro-apoptotic (red) and anti-apoptotic (blue) proteins are shown next to each other. Ensembles of 200–300 cells for each protein antibody were used to estimate the mitochondrial contribution to variability

arises by sampling the initial conditions in protein levels before TRAIL addition from experimental distributions[9]. It was previously described that including correlations between protein pairs along the apoptotic pathway improved the predictive power of the model[27]. These pairwise correlations may be due to direct or indirect interactions, co-regulation by common transcription factors of both proteins or to another common source of gene expression modulation, as we have shown to be the case with mitochondrial content (Fig. 3b, c, Supplementary Methods, Supplementary Sofware 1 and Supplementary Table 3).

A direct way to include the effect of mitochondrial–protein correlations in the model is by sampling from the protein distributions according to the mitochondrial levels. Figure 4a summarises the model workflow. Details of model calibration and parameters are provided in Supplementary Methods and Supplementary Tables 1 and 2. First, we assigned to each cell of the simulated ensemble a random mitochondrial level sampled from the distribution of experimental CMXRos values (Fig. 4b). Then, for each protein in the apoptotic route, we chose a number

of molecules according to a log-normal distribution with mean and standard deviation determined by the particular mitochondria–protein correlation (Fig. 4c and Supplementary Methods). Once the initial conditions have been set for the whole ensemble, we need a criterion to decide the apoptotic fate for each simulated cell. To implement this decision, we noticed that recent experimental results have established that the activation rate of Caspase-8 (Casp8) defines a threshold separating surviving and dead subpopulations[30], which is independent of TRAIL dose. Within our model, we fit this activity threshold to reproduce the probability of death/life at a sensitive dose (32 ng ml$^{-1}$). Those cells whose maximum rate in Casp8 activation overcomes this threshold were considered as apoptotic (Fig. 4d, red lines) while cells below the threshold were considered survivors (Fig. 4d, blue lines). The numerical simulations showed that cells with initially larger mitochondrial levels had higher Casp8 activity rates (Fig. 4e). Finally, the other readout of the model to be compared with experimental data is the time to death for the apoptotic cells. In our model, time to death is taken as the time at which Smac protein reaches 90% of saturated cytosolic levels (Fig. 4f).

Once mitochondrial–protein correlations were included in the model, it qualitatively reproduced all of our experimental findings (Fig. 5). In particular, by fitting the Casp8 activity threshold to reproduce the experimental survival probability at 32 ng ml$^{-1}$ of TRAIL, the simulated dose–response curve followed the trend of the experimental one in the whole range of TRAIL doses, including the sensitive region between 8 and 63 ng ml$^{-1}$ (Figs. 5a and 1b). Similar to the experimental results, the distributions of times to death showed a large spread for low TRAIL doses, while for large doses the majority of cells died within the first 4 h after TRAIL addition, with the exception of a few outliers with long times to death (Figs. 5b and 1c). Model simulations corroborated that mitochondrial levels were able to discriminate cell fate, with AUC values similar to those of the experimental samples (Fig. 5c). Finally, we analysed within our modelling framework the influence of mitochondrial content on apoptosis times. In agreement with the experimental data, cells with low mitochondrial content had systematically longer times to death than cells with high mitochondrial mass (Fig. 5d).

**Mitochondrial heterogeneity and key in variable TRAIL response**. To gain an insight into the effect introduced by mitochondria–protein co-variation on times to death, we simulated two extreme scenarios. In one case, the protein abundance was solely determined by mitochondrial content (correlation coefficient $\rho = 1$), and in the other one, the protein amount was completely independent of mitochondria ($\rho = 0$) (Fig. 6a). The simulation of a perfect mitochondria–protein correlation showed that the times to death followed an inverse non-linear trend with mitochondrial mass, with longest times to death corresponding to cells with low mitochondrial levels (Fig. 6a, red dots). An imperfect correlation between mitochondria and apoptotic proteins scattered times of apoptosis with large deviations around this trend (Fig. 6a, black crosses and grey dots), suggesting that times to death were very sensitive to small changes in protein amount. Therefore, any additional source of protein variability besides mitochondria may also have an impact on the time to death.

To better understand the role of mitochondria–protein co-variation on cell fate, we studied whether the levels of specific proteins of the apoptotic route can discriminate fate with similar accuracy as mitochondrial levels. Hence, we calculated the performance of each apoptotic protein as a classifier of death/life (Fig. 6b, hollow bars). With the exception of Casp8, whose activation rate is used as a discrimination threshold, the

performance of the rest of the proteins in the pathway was worse than that of mitochondria. However, the pro-apoptotic proteins Bid and Bax had discrimination power similar to that of mitochondria. It may be possible that the levels of these two proteins were key determinants of cell death, and the good classification performance of mitochondrial mass arises because of its high co-variation with these specific proteins (Fig. 3c). To investigate this possibility, we repeated the discrimination analysis, sampling protein levels independently of mitochondrial

mass (Fig. 6b, filled bars), showing that no single protein was a proper classifier by itself (AUC < 0.7). These results reinforce the view that mitochondrial mass is an underlying variable that 'predicts' TRAIL-induced apoptosis by its global effect on protein abundance.

We notice that at very low TRAIL doses (4 ng ml$^{-1}$) where receptors are far from being saturated by the ligand, the discriminatory capacity of mitochondria for cell fate seems to improve (Figs. 2b and 5c). Since DR5 receptor levels are correlated to mitochondrial mass (Fig. 3c), it is possible that at these low doses, receptor abundance plays a role in cell death. To test this possibility, we repeated the discrimination analysis shown in Fig. 6b at different TRAIL doses and including only the experimentally observed correlation between mitochondrial and receptor levels (Supplementary Fig. 7). At sensitive and saturating doses (32 and 250 ng ml$^{-1}$), receptor levels have no discriminatory capacity, but at the lowest dose (4 ng ml$^{-1}$), receptor levels are able to partially discriminate cell fate, which may influence the discriminatory capacity of mitochondria.

Finally, we investigated whether cellular fate after the apoptotic stimulus is more sensitive to co-variation of specific proteins with mitochondrial content. We carried out a sensitivity analysis[31] of discrimination performance (AUC) for each protein in the pathway, changing its correlation with mitochondria (Supplementary Methods). As expected, the highest sensitivity corresponds to Casp8, whose activity sets the threshold for cell fate discrimination in the model[9, 30] (Fig. 6c). Moreover, changes in the correlation of the pro-apoptotic proteins Bax and Bid with mitochondria also affected cell fate discrimination, a tighter correlation with mitochondria improved classification performance. Interestingly, their anti-apoptotic partners (Bcl-2 and Mcl-1, respectively) showed negative sensitivity, decreasing classification performance with larger mitochondria–protein correlation. This negative sensitivity was observed in all the anti-apoptotic proteins in the route (Fig. 6c). This sensitivity analysis indicates, on one hand, that if mitochondrial levels are relevant for apoptotic fate, they should exert a tighter control on the abundance of pro-apoptotic proteins, while the levels of anti-apoptotic proteins should be freed from mitochondrial regulation. A higher correlation with mitochondrial mass would effectively couple the abundance of anti-apoptotic proteins to that of the pro-apoptotic proteins, and thus would reduce the discriminating power. On the other hand, it seems that mitochondrial control of protein abundance is especially important for Casp8 and the pre-MOMP pairs of pro- and anti-apoptotic proteins Bid/Mcl-1 and

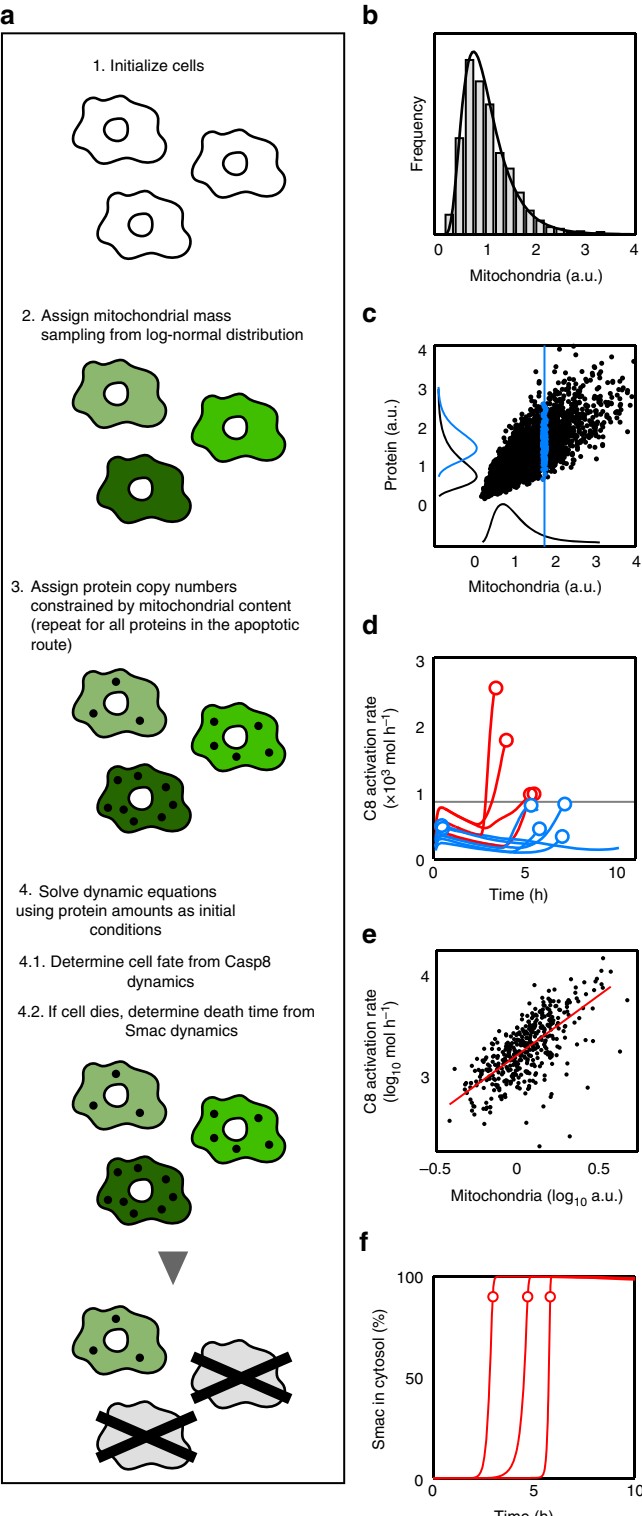

**Fig. 4** Computational model workflow and key modelling aspects. **a** Computational model workflow. **b** Distribution of mitochondrial levels. We initialise a population of cells with heterogeneous mitochondrial levels sampling from a log-normal distribution of mean and width obtained from the experimental CMXRos distribution. **c** Sampling of initial protein levels. For each protein in the apoptotic pathway, we assign to every cell a protein copy number conditioned by its mitochondrial level. The black line along the vertical axis represents the total protein distribution and the blue line, the protein distribution conditioned by the indicated mitochondrial value (vertical blue line). **d** Casp8 activation rate. The decision about the cell fate (death/life) is defined by a threshold (horizontal black line) in the rate of Casp8 activation. Cells with maximum activation rates (circles) below the threshold (blue trajectories) are considered as survivors whereas cells that overcome the activation threshold (red lines and circles) are assumed to die after MOMP. **e** Casp8 maximum activation rate depends on the cell mitochondrial content. Each black dot is a simulated cell (only cells that undergo MOMP before 24 h are shown). The regression line is shown in red. **f** Smac dynamics. Time to death is defined as the time at which cytosolic Smac reaches 90% of its maximum level

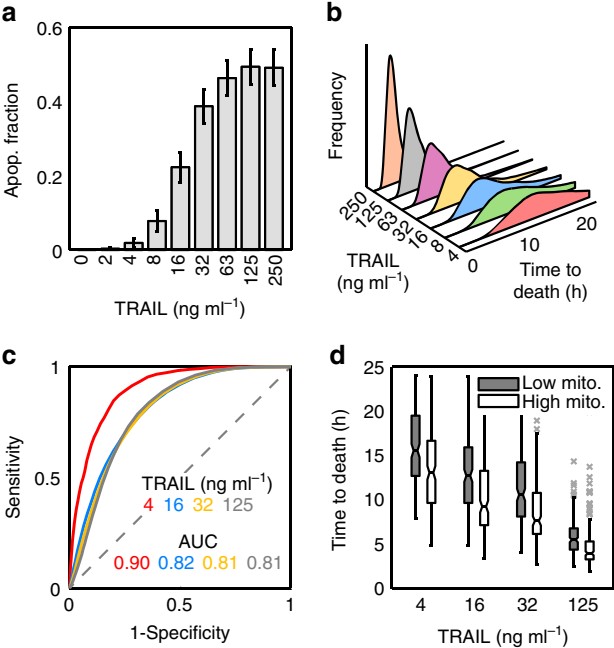

**Fig. 5** Coupling protein variability to mitochondrial mass is enough to explain apoptosis outcome. **a** Fraction of simulated apoptotic cells after 24 h of treatment with TRAIL at the indicated doses. Error bars are computed by bootstrapping. **b** Distributions of apoptosis times at different TRAIL doses from model simulations. **c** Analysis of mitochondrial content as a binary classifier (death/life) of cell fate from model simulations. **d** Boxplots of time to death for simulated HeLa cells with mitochondrial levels in the first quartile (Low mito, grey) and in the fourth quartile (High mito, white) at different doses of TRAIL. Each dose was simulated with an ensemble of 10,000 cells. Boxes cover the range from the lower to the upper quartile of the data. Whiskers indicate maximum and minimum values, excluding outliers which are plotted as individual grey crosses. Horizontal lines inside the boxes represent median values, and notches indicate 95% confidence intervals for the median

Bax/Bcl-2, while its influence on other nodes of the apoptotic pathway may not be so relevant.

Overall, these results suggested that an optimal cell fate discrimination by mitochondria may occur in a regime where co-variation of mitochondria with the pro-apoptotic proteins Bax and Bid is maximal, while their anti-apoptotic partners Bcl-2 and Mcl-1 are de-correlated from mitochondrial mass. Therefore, we calculated the AUC for the two pre-MOMP pairs Bax/Bcl-2 and Bid/Mcl-1 as a function of the MCV at the protein level. We achieved this by sweeping different combinations of mitochondria–protein correlations for the Bax/Bcl-2 or Bid/Mcl-1 pairs while keeping the rest of the correlations at their experimental values. We found that an optimal fate discrimination performance takes place at a high MCV between mitochondria and the pro-apoptotic protein, $\rho_{Bax}$ or $\rho_{Bid}$ close to 1, and low correlation with their anti-apoptotic partners (Fig. 6d). In contrast, discrimination performance substantially decreased for high MCV of the anti-apoptotic proteins with mitochondria, $\rho_{Bcl-2}$ or $\rho_{Mcl-1}$ close to 1. The experimentally measured values were in a regime of close to optimal discrimination (Fig. 6d, black crosses).

**Mitochondrial mass determines apoptotic proteins in tumours.** Our combined experimental and modelling approach has shown that the mitochondrial content predicts apoptotic fate as a consequence of its influence on the levels of apoptotic proteins. We used a clonal population of HeLa cells to eliminate genetic

sources of variability, and homogeneous culture conditions to minimise effects from the microenvironment. In solid tumours, these factors may substantially contribute to variability in resistance to apoptotic drugs. However, we wanted to test whether the same sources of non-genetic heterogeneity in cultured HeLa cells could be found in the individual cells of solid tumours. We followed the same immunolabelling strategy used for HeLa cells to quantify heterogeneity in mitochondrial and protein content, and their correlation. We stained paraffin sections from colon cancer biopsies of three individuals with antibodies against Aconitase 2 as a reporter of mitochondrial mass and, simultaneously, against one of the proteins Bax, Bcl-2, Bid, Mcl-1, Smac, XIAP, Casp8 and Bar (Fig. 7a). These proteins were selected because they constitute pairs of pro- and anti-apoptotic proteins whose correlation with mitochondria had the highest discriminatory power of apoptotic fate in HeLa cells. Similar to the clonal HeLa cell population, tumoural cells from colon cancer exhibit variability in both mitochondria and apoptotic protein levels (Fig. 7b). After calculating the MCV in protein, we also observed a high correlation of mitochondrial mass with the abundance of specific proteins. Moreover, the pro-apoptotic protein in all the protein pairs tested showed higher correlation with mitochondria than the anti-apoptotic partner (Fig. 7c). This result suggests that the mitochondrial content may also determine variability in resistance and apoptotic fate of cells in solid tumours, and constitutes a first step towards the assessment of mitochondrial mass as a biomarker for diagnosis and prognosis in cancer.

## Discussion

A major problem in cancer treatment is chemoresistance. Some chemotherapeutic agents successfully remove most of the bulk tumour but fail to eliminate a minor population of innately drug-resistant cells that continue growing and cause cancer relapse. This variability in behaviour and response is to a large extent due to heterogeneity in the molecular signatures of cancer cells within a tumour. This phenomenon is known as intra-tumoural heterogeneity and can be caused by non-genetic and genetic factors[1, 2, 32–35]. There is increasing evidence that a large fraction of the variability observed at the level of transcripts and proteins in mammalian cells is determined by phenotypic state and population context[7, 36].

The apoptotic pathway is a complex network of proteins involving non-sequential organisation and competing molecular signals that ultimately lead to a binary death/life decision for each single cell[23]. To address this complexity, we adopted a systems level approach involving single-cell experiments and computational modelling. Our experiments revealed that the mitochondrial content discriminates apoptotic cell fate at the single-cell level, and modulates the abundance of all proteins of the apoptotic route, albeit in different ways. To account for the role of mitochondrial content on protein variability, we modified a pre-existing model of the extrinsic apoptotic pathway[27]. Constraining the possible values of the apoptotic proteins by mitochondrial levels, the model reproduced all our experimental observations. Furthermore, our simulations indicate that the power of mitochondrial mass as a death/life classifier depends on differences in tuning specific pro- and anti-apoptotic protein levels. Interestingly, these differences were also observed in cells from colon cancer tumours.

There are biological reasons to single out mitochondrial content, and possibly functionality, as a cellular determinant of programmed cell death. Apoptosis is a physiological process "designed" to eliminate damaged or abnormal cells and to maintain tissue homoeostasis. In that sense, one possibility is that cells with high mitochondrial mass can induce more damage in

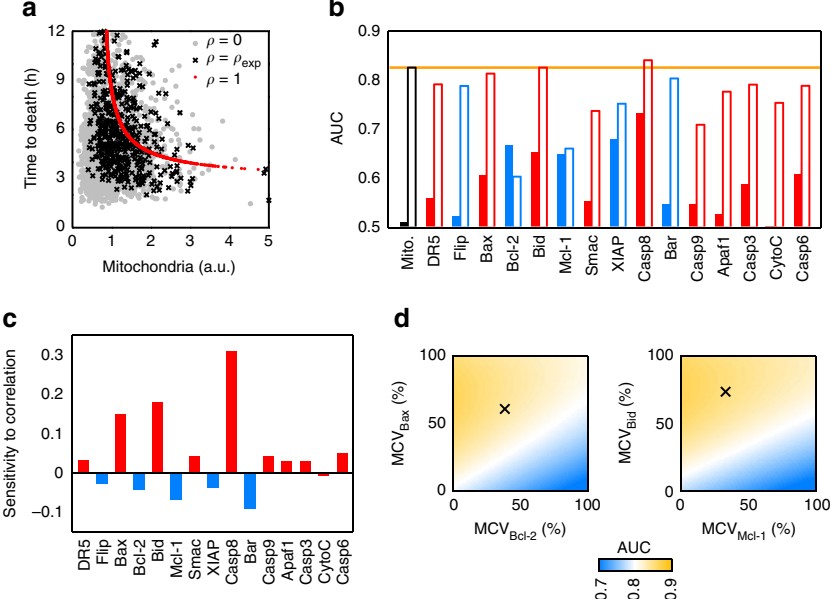

**Fig. 6** Model simulations unveil the effect of protein–mitochondria correlations in the apoptosis outcome. **a** Effect of global modulation of mitochondria–protein correlations on times to death. Red dots: perfect correlation ($\rho = 1$) between mitochondria and protein abundances. Black crosses: experimental correlation values ($\rho = \rho_{exp}$). Grey dots: no correlation ($\rho = 0$). **b** Performance of apoptosis fate discrimination using each apoptotic protein as a binary classifier. Red bars correspond to pro-apoptotic proteins and blue bars to anti-apoptotic proteins. Hollow bars: including experimental mitochondria–protein correlations. Filled bars: protein variability sampled independently of mitochondrial levels. The discriminatory performance of mitochondria is included in orange as a reference. **c** Sensitivity of AUC to changes in individual mitochondria–protein correlations. Negative sensitivity corresponds to a situation where increasing a particular mitochondria–protein correlation decreases the discrimination performance. **d** Discrimination performance as a function of the mitochondrial contribution to protein variability (MCV) for the two pre-MOMP pairs of pro- and anti-apoptotic proteins Bax/Bcl-2 (left panel) and Bid/Mcl-1 (right panel). The black crosses indicate the experimental value of both MCVs. All model simulations in this Figure have been carried out at a sensitive dose (32 ng ml$^{-1}$). Ensembles of 10,000 cells were used for each simulation

DNA than cells with low mitochondria, because mitochondria are the major cellular source of reactive oxygen species (ROS). In line with this, ROS levels scale with mitochondrial content in HeLa cells (Supplementary Fig. 8a). However, there is also a strong linear correlation between antioxidant levels and mitochondrial abundance (Supplementary Fig. 8b). This suggests that although cells with more mitochondria produce more ROS, it is balanced by the cell's thiol defence (Supplementary Fig. 8c and d). For that reason, when HeLa cells were exposed to NAC, in order to build more antioxidant defences, we did not observe changes in the death index (Supplementary Fig. 8e). However, when the anti-oxidant defences (thiols) of the cell are removed by diamide, apoptotic susceptibility to TRAIL increases until all cells are killed. These facts strongly suggest that the higher cell death of cells with high mitochondria is not due to these cells being exposed to more ROS than low mitochondria cells.

Alternatively, mitochondria also modulate the ratios of many metabolites such as ATP/ADP, acetyl-CoA/CoA, NAD+/NADH and NADP+/NADPH, which can act as metabolic checkpoints for cell death[37]. It may be the case that cells with larger mito-chondrial mass have a metabolic status with more imbalanced metabolite ratios, that prime them for death after a severe stress. The results presented here suggest that global metabolic control of programmed cell death is achieved in a 'passive' way by exploiting the fact that mitochondrial content modulates protein abundance.

Mitochondrial levels contribute differently to the variability of the proteins in the apoptotic pathway, with stronger control on the abundance of specific pro-apoptotic proteins that are key for triggering MOMP and activation of Casp8, but weak co-variation with their anti-apoptotic partners. Modelling indicates that this dependence of key pro- and anti-apoptotic proteins confers

mitochondria a high discriminatory capacity for apoptotic fate. Supporting this finding, apoptotic susceptibility can be deter-mined by the levels of pre-MOMP pro-apoptotic proteins of the BH3 family[38] while the levels of anti-apoptotic proteins are considered as a 'buffer' to protect cells against basal levels of pro-death signals that are encountered in normal physiological con-ditions[39]. In line with this, tumours with cells in which pro-apoptotic signalling was highly primed showed a better clinical response to different chemotherapeutic agents[40].

There is more evidence in the literature connecting mito-chondrial mass and functionality to apoptotic fate and response to chemotherapy. For instance, leukaemia cells have been found to have a larger mitochondrial mass, a greater mitochondrial DNA copy number and a higher rate of oxygen consumption than normal hematopoietic cells, and were selectively killed by drugs inhibiting mitochondrial protein synthesis[41]. On the other hand, downregulating mitochondrial function by retrograde sig-nalling promotes endothelial–mesenchymal transition (EMT)[42], which is linked to metastasis[43]. Moreover, metastatic cells are chemoresistant[44]. Recently, the transcriptomic analysis of 20 different types of cancers (8161 cancer and normal samples) has shown that the downregulation of mitochondrial genes was associated with the worst clinical outcome and correlates with the expression of genes promoting metastasis across many cancer types[45].

Overall, our results suggest that mitochondrial content is a good biomarker for the prediction of apoptotic susceptibility. Despite the overwhelming amount of studies, in some cancer types such as colorectal tumours, no optimal biomarkers have been described[46]. To remedy this situation, some authors have proposed the combined use of the whole apoptotic profile of a tumour, rather than the expression of single markers, to improve

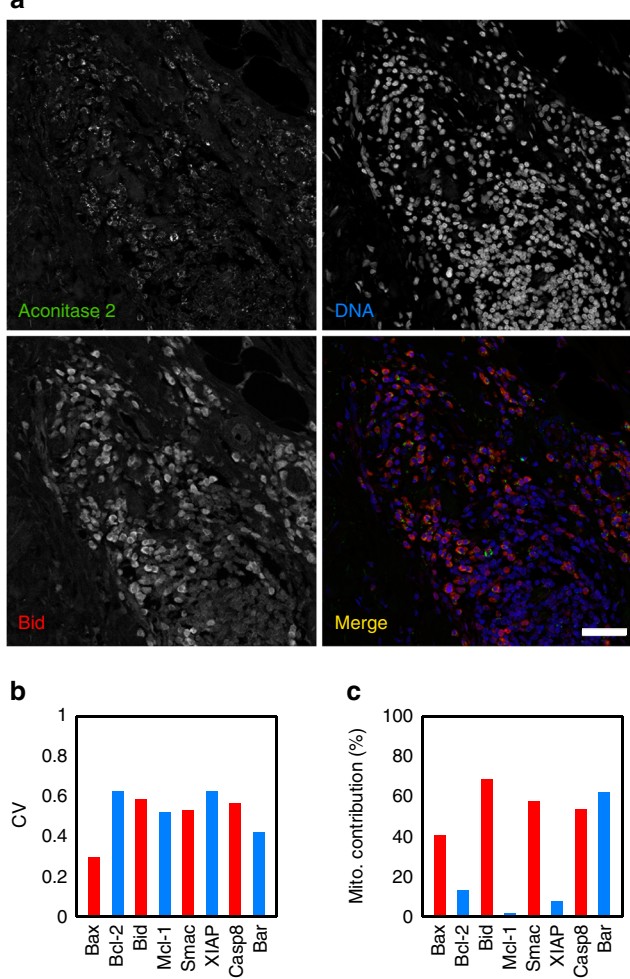

**Fig. 7** Colon cancer cells show variability in apoptotic proteins and mitochondria. **a** Colon cancer section stained with Aconitase 2, Bid and DAPI. This image illustrates the variability in expression of Bid and Aconitase 2. Scale bar: 50 μm. **b** Coefficient of variation (CV) of several proteins involved in the apoptosis pathway. **c** Mitochondrial contribution to variability of the apoptotic proteins. Data are representative of four independent biopsies. Statistical quantities were obtained from ensembles of 500–1000 cells for each protein antibody

the prognosis and treatment of cancer patients[46]. The observed changes in the expression of pro-apoptotic genes with mitochondrial levels in colon cancer samples raise the possibility to implement the amount of mitochondria as a unique biomarker, representing the final outcome of the whole apoptotic pathway. The validation of this surmise will require an extensive analysis of different cancer samples and chemotherapeutic drugs, together with their clinical response.

## Methods

**Cell lines and materials**. HeLa (ATCC CCL-2) cells were grown in Dulbecco's Modified Eagle Medium (DMEM, Gibco)–GlutaMAX-I supplemented with 10% foetal bovine serum (FBS, Hyclone) and penicillin–streptomycin (Sigma) in a 37 °C humidified incubator with ~5% $CO_2$. Mitochondrial mass for in vivo experiments was measured as the integrated signal of MitoTracker Green FM (MG, Molecular Probes) incorporated by individual cells. In fixed cells, mitochondrial mass was measured using MitoTracker Red CMXRos (CMXRos, Molecular Probes). The apoptotic signal was triggered by human recombinant TRAIL (Millipore) at the indicated dosses.

**TRAIL apoptosis assay in HeLa cells**. HeLa cells were seeded in 24-well plates (Nunc) and incubated with increasing doses from 2 to 250 ng ml$^{-1}$ of TRAIL for 24 h. After the treatment, both the dead-suspended cells and the live-adherent cells were collected. Then, the cells were washed twice with PBS and stained with Annexin V-FITC/PI (Propidium Iodide). Apoptotic analysis was performed using a FACSCalibur flow cytometer.

**Live cell microscopy**. HeLa cells were seeded in 24-well plates (Falcon) 1 day before the experiments. Prior to addition of apoptotic inducers, the cells were stained for 40 min with MG and washed twice with DMEM; 15–30 min prior to the start of the movie, the cells were added to the culture medium: TRAIL, at the appropriate dilution, or 63 ng ml$^{-1}$ of DRB, or 2.5 μg ml$^{-1}$ of CHX or a combination of 2.5 μg ml$^{-1}$ of CHX plus TNF at 20 ng ml$^{-1}$. HeLa-treated cells were imaged at 15-min intervals for 24 h in a 37 °C humidified chamber in ~5% $CO_2$. The cells were imaged at 20× magnification (0.4 NA HCX PL FL) on a Leica DMi6000b microscope (Leica MicroSystem) equipped with a Hamamatsu Orca-R2 digital CCD Camera and the images were acquired using the LAS AF 2.7 software (Leica MicroSystem). Time to death was monitored by morphological changes associated with apoptosis. The images were analysed using Fiji 2.0.0-rc-43 software[47]. The mitochondrial level was calculated from the first fluorescence image, and the cell fate by morphological changes associated with apoptosis, at the end of the experiment.

**Effect of ROS in TRAIL induced apoptosis**. HeLa cells were seeded in 24-well plates 1 day before the experiment. Initially, the cells were stained for 40 min with MG and washed twice with DMEM. Secondly, the cells were treated for 2 h with Diamide (20, 50 and 100 μM) or N-acetyl cysteine (NAC) (0.5, 1 and 2 mM). Third, TRAIL was added to the culture medium at a concentration of 32 ng ml$^{-1}$. Finally, the live-cell microscopy experiments were performed imaging at 15 min intervals for 24 h in a 37 °C humidified chamber in ~5% $CO_2$.

**Mitochondrial ROS and cellular antioxidant defence measure**. To analyse the mitochondrial ROS production according to the levels of mitochondria, HeLa cells growing on coverslips were stained with MitoSOX (Molecular Probes) for 1 h and with MG for the last 20 min of the MitoSOX incubation. The coverslips were washed twice with PBS and then the cells were fixed. The coverslips were mounted in Vectashield.

The cellular antioxidant defence was studied by measuring the reduced thiol group with ThiolTracker (Molecular Probes). To this end, HeLa cells growing on coverslips were stained for 20 min with CMXRos. Then, the cells were washed twice with PBS and fixed. The fixed cells were stained for 15 min with ThiolTracker, washed twice with PBS and mounted in Vectashield.

**Immunostaining using wide confocal cytometry**. HeLa cells growing on coverslips were fixed and proteins indirectly immunolabelled using the corresponding primary antibodies. Secondary antibodies were Alexa Fluor 488, 546 or 647 donkey anti-mouse, goat or rabbit IgG (H+L) (Invitrogen). The coverslips or slides were mounted in Vectashield (Vector Laboratories). The images of the labelled cells were collected in a Leica TCS Sp5 multispectral confocal system (Leica MicroSystem), with a 20× 0.7 HCX PL APO CS, with the pinhole completely opened in order to collect the maximum amount of light emitted by the specimen. Hundreds of cells in different fields of the slide were collected. These images were exported to and analysed with MetaMorph 7.8.0.0 software (Molecular Devices).

For colon cancer immuno-histochemistry, tumour biopsies were formalin fixed and paraffin embedded. Tissue sections (5 μm) were treated with EnVision FLEX Target retrieval solution low pH (DAKO) (95 °C, 2 min) in order to unmask the antigens. The immunolabelling was performed in the same way as that for the cultured cells.

The protein antibodies were used at dilution 1:1000 and were purchased from Abcam: Flip (ab167409), XIAP (ab137392), Aconitase 2 (ab110321 and ab99467), Bar (ab106547) and DR5 (ab8416); Santa Cruz Biotechnology: Bak (sc832) and Bax (sc493); Cell Signaling Technology: Smac/Diablo (15,108); Cusabio Biotech: Bcl-10 (CSB-PA002608ESR2HU); and from Sigma Prestige Antibodies: Casp8 (HPA005688), Casp9 (HPA001473), Bcl-2 (B3170) and Mcl-1 (HPA008455).

**RNAseq and data processing**. HeLa cells were stained with MG for 40 min in DMEM. After the staining, the cells were washed twice with PBS, trypsinized and resuspended in PBS with 5 mM EDTA. Then, the cells were sorted on a fluorescence-activated cell sorter MoFlo XDP (Beckman Coulter) into two populations of $10^6$ cells with high and low mitochondrial content with a difference in mitochondrial mass of around 5-fold. Following the DNase treatment, total RNA from sorted cells was extracted using RNeasy Mini Kit (QIAGEN) according to the manufacturer's guidelines. The quality of the extracted RNA was measured by RNA Integrity Number (RIN) value from Bioanalyzer (being in all cases higher than 8), and finally, 3 μg of purified RNA was sent to RNA-sequencing at the SNP&SEQ sequencing facility (Science for Life laboratory (SciLifeLab), Uppsala sequencing node). Total RNA was depleted from rRNA prior to library construction. One lane per sample was used in a 60-bp paired-end run on an Illumina

HiSeq 2500 sequencer. For each sample, over 50 million pair-end reads were sequenced.

In order to quantify the total amount of mRNA per individual cell of each population, sorted HeLa cells were seeded into cover-slide. To quantify mRNA carrying poly(A), we performed an RNA-FISH detection experiment using poly-T-Alexa-488 following the protocol described in Brown and Buckle[48]. Briefly, HeLa cells growing on coverslips were fixed for 10 min with PFA 4% (Electron Microscopy Science) and permeabilized with 0.5% TRITON X-100 (Sigma) for 6 min at 4 °C. Then, the cells were washed twice in SSC 2×. The cells were labelled with 100 ng of poly-T in 12 μl of RNA HM solution (25% formamide, 200 ng/μl yeast tRNA, 5× Denhardt Solution, 1 mM EDTA, 2× SSC) over night at 37 °C in a humidity chamber. Finally, the coverslips were washed three times with 2× SSC at 37 °C, labelled with DAPI and mounted in Vectashield.

Sequenced reads were aligned to the *Homo sapiens* genome (version GRCh38 from Ensembl) using TopHat 2.1.1[49] linked to Bowtie 2.2.8[50] with default sensitive settings. From sequenced reads, the transcripts were assembled using Cufflinks 2.2.1[51] using the Ensembl GRCh38.84 annotation as reference. Transcript/gene abundances were estimated using Cufflink's standard unit of measurement (fragments per kilobase per million reads, FPKM) and then transformed into transcripts per million (TPM)[52]. TPM values were corrected in the 'low' sample by a factor 0.34 to account for the different per-cell RNA content in the 'low' and 'high' conditions, as quantified by poly(A) RNA-FISH. Additional information for each gene was obtained from the Ensembl BioMart database and included in the data set.

We then discarded the genes whose expression level was below the detection threshold in both 'low' and 'high' conditions. This threshold was estimated as follows[25]: all genes with one zero and one non-zero expression value in any condition were selected. All non-zero values of this set of genes were listed, and the detection threshold calculated as the median of their distribution, which was 0.01 TPM. Expression levels below this cutoff value in the filtered data set were replaced by the cutoff.

**Human colorectal samples**. Human colorectal tumour biopsies from de-identified patients were obtained with signed patient-informed consent and approval from the Human Ethics Review Committee of the Torrevieja and Vinalopó Hospitals.

**Data availability**. The authors declare that all data supporting the findings of this study are available within the article and its supplementary information files or from the corresponding author upon reasonable request. RNAseq data have been deposited in the NCBI database under accession code BioProject ID: PRJNA416451. Simulation codes are written in Matlab (Mathworks 2015a) and are available as Supplementary software files.

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

## Acknowledgements

The Spanish Ministry of Economy and Competitiveness (MINECO) supported this research under grants BFU2013-45918-R and BFU2016-79127-R. F.J.I. acknowledges a grant from the European Sequencing and Genotyping Infrastructure (ESGI), Grant Agreement no. 262055. J.D.C. is a recipient of a Ph.D. fellowship 'Severo Ochoa' Excellence Program from MINECO. R.G. acknowledges funding from the AIRBIOTA-CM project of Comunidad Autónoma de Madrid (S2013/MAE-2874). R.G. and J.D.C. acknowledge the Scientific Computing Center at Universidad Autónoma de Madrid (CCCUAM) for access to their computing facilities.

## Author contributions

F.J.I. and R.G. designed and supervised the research; S.M.-J. and F.J.I. conducted the experiments; J.D.-C. and R.G. developed the computational model and performed statistical analysis; J.D.-C. conducted the numerical simulations and model fitting; J.D.-C., S.M.-J., R.G. and F.J.I. analysed the data; R.P.d.N. conducted the preliminary experiments; A.M.-L. provided the colon cancer samples; F.A. contributed to discussions; F.J.I. and R.G. wrote the paper, with contributions from J.D-C, S.M.-J and F.A. All authors read and approved the final version of the manuscript.

## Additional information

**Competing interests:** The authors declare no competing financial interests.

