## [Peer Review File · Nature Communications]

Reviewers' Comments:

Reviewer #1:

Remarks to the Author:

In this manuscript, the authors study the impact of heterogeneity in mitochondrial content on the outcome of TRAIL-induced apoptosis in HeLa cells. The authors show as the cellular mitochondrial content determines the apoptotic fate and modulates the time to death, by the control of apoptotic protein abundance, in particular, pro-apoptotic proteins Bax and Bid. Furthermore, they found a strong correlation between mitochondrial content and apoptotic proteins levels was also observed in colon cancer biopsies, suggesting that mitochondrial mass is a good prognosis biomarker.

The statistical analysis is excellent and well performed but limited to a specific kind of cell or tumor type. The author use TRAIL to induce apoptosis. Is the proposed model valid also for other apoptotic inducers? Or in different cell types?

The authors affirm that the increased sensibility to TRAIL treatment is strictly related to the amount of mitochondria and mitochondrial apoptotic proteins, but they can't exclude that this effect could be due to heterogeneous expression of the receptors, activated by the apoptotic stimulus TRAIL on the plasma membrane, among the different cells. Indeed, increased levels of these receptors could explain the augmented activation of the apoptotic pathway.

As described in PMID: 26538029, Mcl-1 exists in two different forms, the pro-apoptotic Mcl-1 L, and the anti-apoptotic Mcl-1 S

and the ratio Mcl-1S/Mcl-1L is determinant for the sensitivity of cancer cells to apoptotic stimuli via regulation of mitochondrial homeostasis.

The authors refer only to Mcl-1 anti-apoptotic activity, but they do not specified if they evaluate the total amount of the protein or only the pro-apoptotic Mcl-1 L.

Minor point:

In general, increase the size of the graphs shown for a better comprehension.

Fig. 1B- fig.5A: please indicate all the TRAIL doses used for the experiment.

Reviewer #2:

Remarks to the Author:

In this manuscript, Diaz-Colunga and colleagues investigate mechanism(s) underlying fractional killing by Trail. As the authors state, fractional killing by anti-cancer therapies is a major source of disease recurrence. Using HeLa cells as a paradigm, the authors find that mitochondrial content correlates with sensitivity to Trail-mediated killing - cells with higher mitochondrial content are more sensitive to Trail treatment. Building on their earlier work, the authors find higher transcript levels and in particular a correlation with pro-apoptotic BCL-2 proteins in high-mito content cells. In my opinion (with the possible caveat described below), the authors convincingly demonstrate a correlation between mitochondrial content and apoptotic sensitivity. However, as discussed below, there is no mechanistic insight underlying this correlation. In my opinion, it remains formally unproven whether higher mitochondrial content is a cause or consequence of increased apoptotic sensitivity. Points to consider are below.

- further investigation into why higher mitochondrial content correlates with higher apoptotic sensitivity is required. The authors suggest that this may be due to increased BAX/BID levels for example, however I am somewhat sceptical given that these proteins on a per cell basis are in significant excess for what is actually required to induce MOMP - nevertheless, this is a testable hypothesis, for example does increased BID or BAX expression ablate the mitochondrial content/apoptosis sensitivity correlation

- Alternatively, as the authors discuss increased ROS caused by increased mitochondrial content

may also underlie the increased apoptotic sensitivity, again this is testable.

- given the reliance on it, an initial (hopefully) confirmation that mitotracker green is indeed measuring mitochondrial mass using a couple of different parameters (e.g. additional mitochondrial protein stain and/or mtDNA content) is important to show.

Reviewer #3:

Remarks to the Author:

In their manuscript, Diaz-Colunga and Márquez-Jurado and colleagues tackle the fundamentally important and clinically relevant problem of non-genetic heterogeneity in cellular responses to apoptotic stimuli. They show experimentally that in HeLa cells, the mitochondrial mass in a cell correlates with both its the TRAIL-induced cell fate as well as its abundance of proteins involved in the apoptotic response to TRAIL. Computationally, they show that recapitulating the distribution of mitochondrial mass and its correlation with the expression of regulators of extrinsic apoptosis is sufficient to recapitulate several features of the cellular response to TRAIL and its heterogeneity. Finally, the authors show that similar correlations between mitochondrial mass and the abundance of apoptotic regulator also exist in cells from colon cancer samples. Overall, this is a well written manuscript that addresses the important problem of fractional killing in TRAIL-treated cancer cells using well thought-out analyses based on solid data. The authors should address a few concerns to further strengthen their manuscript.

Major issues

1. With regards to the simulations of the EARM model, there is one central assumption by this work that is glossed over in the manuscript: that the mitochondrial mass is equivalent to the concept of 'Pores'. Because of the central role of 'pores' in the initiation of the MOMP pathway, the authors should more explicitly analyze and discuss how much this assumption is central to their computational findings.
2. With regards to the correlation of measured mitochondrial mass with cell fate and time to death (fig. 2), one alternative explanation for the data is that there is a certain toxicity from MitoTracker Green (and/or exposure to light in the presence of the dye), and that cells with more dye are more likely to die, and die early. Are the distributions of cell death times and the fraction of apoptotic cells affected by the use of MitoTracker Green?

Other issues

3. p.6 – with regards to the validation of the long half-life of TRAIL in the culture supernatant, the authors cite 'data not shown', however, I believe the policy of Nature journals is to request that all supporting data be presented in main or supplementary figures.
4. p.7 – with regards to the discussion of the influence of the cell cycle on cell death, the analysis in supp. fig. 2c is well done, but the authors should clearly state how the first analysis in sup. fig. 2a is flawed and thus requires the simulation test done in supp. fig. 2c: in the way the data is collected, there must be a bias as observed in part 2a, as a cell certainly cannot have a long time to death if it has a short time to division.
5. p.11 – with regards to evaluating time of cell death in simulation results, I assume that the time at which Smac protein reach 90% of 'saturated' cytosolic levels is evaluated only for cells which are deemed apoptotic? Are similar 'saturated' cytosolic levels reached in all cells called apoptotic? If not, how is time of death impacted by this particular definition?

6. p.12-3 – it should be made clearer that when performing the discrimination analysis for simulations accounting for co-variation with mitochondrial mass, Bid and Bax abundance, which have some discrimination power, are **not** independent from the abundance of other proteins in the system, while when sampling is done independently, they are. Therefore, the discrimination power of Bid and Bax is likely linked to the fact that they can ‘predict’ the abundance of several proteins in the system.

7. p. 13 – it should be made clear that the negative sensitivity of discrimination performance for Bcl-2 and Mcl-1 does not necessarily reflect that they are not important to the cell fate, but rather that a higher correlation with mitochondrial protein mass effectively couples their abundance to that of pro-apoptotic proteins, and thus reduces discriminating power.

8. p.17 (lines 403-412) – the various connections between mitochondrial mass, mitochondrial function and mitochondrial gene expression and certain cell fates are not necessarily support for the correlation between mitochondrial mass and propensity for apoptosis, but rather interesting and important contexts in which this connection certainly merits investigation.

9. p.17 – while the data presented in the manuscript does suggest that mitochondrial content could be a good biomarker for sensitivity to TRAIL, but as TRAIL itself is not at this time a clinically relevant cell death stimulus, the connection should be demonstrated for other pro-apoptotic stimuli or clinically relevant anti-cancer drugs before an extensive analysis is done in cancer samples, as suggested.

Reviewer #1 (Remarks to the Author):

In this manuscript, the authors study the impact of heterogeneity in mitochondrial content on the outcome of TRAIL-induced apoptosis in HeLa cells. The authors show as the cellular mitochondrial content determines the apoptotic fate and modulates the time to death, by the control of apoptotic protein abundance, in particular, pro-apoptotic proteins Bax and Bid. Furthermore, they found a strong correlation between mitochondrial content and apoptotic proteins levels was also observed in colon cancer biopsies, suggesting that mitochondrial mass is a good prognosis biomarker.

The statistical analysis is excellent and well performed but limited to a specific kind of cell or tumor type. The author use TRAIL to induce apoptosis. Is the proposed model valid also for other apoptotic inducers? Or in different cell types?

We thank the reviewer for her/his positive comments on the analysis and for the suggestions made that have helped to clarify some issues and improve our original manuscript.

Following the referee advise we have tested other apoptosis inducers, one from the extrinsic program (TNF) and two for the intrinsic one (DRB and Cycloheximide). In all the tested cases mitochondria can discriminate cells undergoing apoptosis. This data has been included in the Supplementary Information and presented below.

The authors affirm that the increased sensibility to TRAIL treatment is strictly related to the amount of mitochondria and mitochondrial apoptotic proteins, but they can't exclude that this effect could be due to heterogeneous expression of the receptors, activated by the apoptotic stimulus TRAIL on the plasma membrane, among the different cells. Indeed, increased levels of these receptors could explain the augmented activation of the apoptotic pathway.

While it is true that mitochondrial mass is correlated with the amount of DR5 receptors (Figure 3c, around 55% of the cell-to-cell variability in DR5 levels is explained by mitochondrial content, see also figure attached), our results pointed out to a small role of receptor number on apoptotic fate (for a fixed TRAIL dose). First, from Figure 2b, we notice that both at a low TRAIL dose (4 ng ml^{-1}) where receptors are far from being saturated by TRAIL ligands, and at saturating TRAIL doses (125 ng ml^{-1}) the discriminatory capacity of mitochondria is similar. Second, model simulations suggest that, at sensitive TRAIL doses, the ability of DR5 to discriminate apoptotic fate per se is low (Figure 6b), and mitochondrial discriminatory capacity is not very influenced by correlation with DR5 (Figure 6c).

Nevertheless, we observed that discrimination of apoptotic fate by mitochondrial content seems to improve at very low ligand doses (4 ng ml^{-1}) both in experiments and model simulations (Figure 2b and 5c), and at these doses receptor levels may play a role in cell death. To further investigate this point as the referee suggests, we repeated the discrimination analysis shown in Figure 6b, at different TRAIL doses and compared a situation including ALL correlations of mitochondrial mass with proteins and receptors (attached figure, empty bars) with a case in which ONLY the correlation between mitochondrial and receptor levels has been included (attached figure, filled bars). We see that at intermediate/high doses (32 and 250 ng ml^{-1}) receptor levels have no discriminatory capacity, but at the lowest dose (4 ng ml^{-1}), receptor levels are able to partially discriminate cell fate, which influence the discriminatory capacity of mitochondria. These results could explain the improved ability of mitochondria to discriminate cell fate observed at 4 ng ml^{-1} . We have included this analysis in the main text and added this figure as Supplementary Fig. 7.

As described in PMID: 26538029, Mcl-1 exists in two different forms, the pro-apoptotic Mcl-1 L, and the anti-apoptotic Mcl-1 S and the ratio Mcl-1S/Mcl-1L is determinant for the sensitivity of cancer cells to apoptotic stimuli via regulation of mitochondrial homeostasis.

The authors refer only to Mcl-1 anti-apoptotic activity, but they do not specified if they evaluate the total amount of the protein or only the pro-apoptotic Mcl-1 L.

The referee is right, Mcl-1 exist as two major alternative splice forms, with the long form being anti-apoptotic and the short pro-apoptotic. Despite our antibody is able, according to manufacturer, to recognise both forms, the long form is the only present in our clone of HeLa cells, as we show by western blot (Sigma prestige HPA008455). For western-blot, HeLa cells were sorted according to mitochondria content as described in

Guantes et al., 2015. For loading control a protein of the mitochondrial matrix Aconitase 2 was used.

Moreover, our transcriptomic analysis also substantiates this finding.

Name	TPM High	TPM Low	P val	Gene id	Transcript id	Protein id
Mcl-1 L	154.68	46.60	0.06	ENSG00000143384	ENST00000369026	ENSP00000358022
Mcl-1 S	4.04	1.20	0.04	ENSG00000143384	ENST00000307940	ENSP00000309973

Minor point:

In general, increase the size of the graphs shown for a better comprehension.

Fig. 1B- fig.5A: please indicate all the TRAIL doses used for the experiment.

We have added the missing labels in Fig. 1b, 5a. In the case of the figure size, we apologize if the size is not big enough, our figures are adjusted to the final width that the journal recommend.

Reviewer #2 (Remarks to the Author):

In this manuscript, Diaz-Colunga and colleagues investigate mechanism(s) underlying fractional killing by Trail. As the authors state, fractional killing by anti-cancer therapies is a major source of disease recurrence. Using HeLa cells as a paradigm, the authors find that mitochondrial content correlates with sensitivity to Trail-mediated killing - cells with higher mitochondrial content are more sensitive to Trail treatment. Building on their earlier work, the authors find higher transcript levels and in particular a correlation with pro-apoptotic BCL-2 proteins in high-mito content cells. In my opinion (with the possible caveat described below), the authors convincingly demonstrate a correlation between mitochondrial content and apoptotic sensitivity. However, as discussed below, there is no mechanistic insight underlying this correlation. In my opinion, it remains formally unproven whether higher mitochondrial content is a cause or consequence of increased apoptotic sensitivity. Points to consider are below.

We thank the reviewer for her/his comments on the reach of our conclusions and for suggesting new experiments and validations, which have helped to clarify some issues.

Regarding the mechanistic link between mitochondria and apoptotic proteins, our group has demonstrated in a series of papers (das Neves et al., PLoS Biology 2010; Johnston et al., PLoS Computational Biology 2012; Guantes et al., Genome Research 2015; Guantes et al., BioEssays 2016) the relationship between mitochondria and the biosynthetic capabilities of the cell. Mitochondria content, through its role in ATP production, modulates RNA pol II activity, being more active and producing more mRNAs in cells with more mitochondria, and the same scenario applies to translation. For that reason, the more plausible explanation for the connection between mitochondria and apoptotic proteins is a global effect of mitochondria on the biosynthesis of these proteins.

Apoptosis is a process thoroughly studied which has led to a general well-established mechanism, for both extrinsic and intrinsic pathways. Using the apoptotic network and a kinetic model previously tested in similar systems, we are able to explain our experimental results considering only the experimentally obtained correlations between mitochondrial mass and the abundance of the main proteins of the apoptotic route. Therefore, we are not describing a new apoptotic mechanism, but only the underlying regulation of the route in individual cells.

Since over and under-expression experiments usually affect many aspects of cell physiology, we resorted to the computational model to gain insight into the causes of fractional cell killing in a non-distorted scenario. In this sense, model simulations (Figure 6b) support the view that mitochondrial mass determine apoptotic fate modulating the initial levels of all pro and anti-apoptotic proteins in the pathway. We found a special sensitivity to correlation with mitochondria in the cases of the pro-apoptotic proteins Bax and Bid (Figure 6c). However, we think that what is relevant for mitochondrial discrimination of apoptotic fate is the differential correlation between the pro-apoptotic proteins and their corresponding anti-apoptotic partners (Bcl-2 and Mcl-1 in this case), as we argue with the simulations in Figure 6d. We have emphasized this aspect in the revised version of our manuscript, and changed the abstract to avoid confusion with respect to the role of Bax or Bid only.

- further investigation into why higher mitochondrial content correlates with higher apoptotic sensitivity is required. The authors suggest that this may be due to increased BAX/BID levels for example, however I am somewhat sceptical given that these proteins on a per cell basis are in significant excess for what is actually required to induce MOMP - nevertheless, this is a testable hypothesis, for example does increased BID or BAX expression ablate the mitochondrial content/apoptosis sensitivity correlation.

While it is true that low levels of activated Bax are required to induce MOMP (Düssmann, Rehm et al., Cell Death and Differentiation (2010) 17, 278–290) we think that it is the balance between the levels of this protein and other pro and anti-apoptotic

proteins which ultimately decides cell death (J. E. Chipuk and D.R. Green, Trends in Cell Biology (2008) 18, 157). According to the comments of the referee, if Bax was in excess, an increase in the levels of Bax should not have impact on apoptosis induction after TRAIL treatment.

We tested such a possibility. We have overexpressed Bax in HeLa cells using a plasmid expressing the human version of Bax under the Cytomegalovirus promoter (see western blot). We observed that in cells with overexpressed Bax, the fraction of apoptotic cells increased by 45% and 30% after exposure at 16 and 32 ng ml⁻¹ of TRAIL respectively. This indicates that in normal cells apoptosis is still susceptible to the levels of inactive Bax. This finding is also in agreement with the view that it is the balance between the levels of this protein and other pro and anti-apoptotic proteins which ultimately decides cell death.

In this figure HeLa stands for HeLa cells non-transfected, Neg are HeLa cells transfected with a mock plasmid and Hyper Bax are HeLa cells transfected with Bax. In the panel below we show the loading control histone H2B.

Then, we checked in hyper-expressing cells the correlation between Bax and mitochondria that was 0.68 vs 0.8 in control cells (Figure below, panel a). Next, we checked for the discriminatory role of mitochondria in hyper-expressing Bax cells. The box plot analysis showed that mitochondria were good discriminators of cells that undergo apoptosis (Figure below, panel b). When our experimental (hyper-expressing Bax cells) data on protein expression and correlation with mitochondria were introduced in the mathematical model, it was able to reproduce the experimental increase in mortality and the correlation between mitochondria content and apoptotic fate (Figure below, panel c and d).

The reason for such a phenomenon was that Bax hyper-expressing cells maintain the correlation of expression with mitochondria, because mitochondria are global regulator of gene expression (das Neves et al, 2010 and Guantes et al., 2015).

The new data reinforce our conclusion where the important thing for the discriminatory power of mitochondria is the differential correlation between mitochondria and pro and anti-apoptotic proteins.

- Alternatively, as the authors discuss increased ROS caused by increased mitochondrial content may also underlie the increased apoptotic sensitivity, again this is testable.

We have performed new experiments to substantiate this hypothesis, Supplementary Fig. 8 in the revised version of the manuscript (also attached below). We quantified simultaneously mitochondrial superoxide levels (using MitoSOX) and mitochondrial mass (MitoTracker Green), and global anti-oxidant cell levels (ThiolTracker) as a function of another mitochondrial mass reporter (CMXRos). We observed that there is a positive linear correlation between both pro-oxidants and anti-oxidant levels and mitochondrial abundance. This suggests that although cells with more mitochondria produce more ROS, this is balanced by the cell's thiol defence. To evaluate the cell's buffering capacity for mitochondrial ROS, we treated HeLa cells with the thiol-oxidizing agent diamide, or with the antioxidant N-acetyl cysteine (NAC), before adding TRAIL at a sensitive dose (32 ng ml^{-1}). We observed that the treatments by themselves don't trigger apoptosis. But, when TRAIL is added, cells treated with NAC showed no significant difference in apoptosis with control cells (magenta bars). However, when the anti-oxidant defences of the cell are removed by diamide, apoptotic susceptibility to TRAIL increases until all cells are killed (orange bars). This indicates that, although cells are certainly sensitive to ROS levels when subjected to an

apoptotic trigger, the anti-oxidant pool is capable to buffer excess mitochondrial ROS in normal conditions. Therefore, we rule out, at least in our experimental system, the role of ROS in the observed apoptotic susceptibility of cells with more mitochondria.

We have changed the discussion section to include these new observations.

- given the reliance on it, an initial (hopefully) confirmation that mitotracker green is indeed measuring mitochondrial mass using a couple of different paramaters (e.g. additional mitochondrial protein stain and/or mtDNA content) is important to show.

We understand the referee's concern. For that reason, we addressed this problem in our previous paper, published in Genome Research in 2015 (Guantes et al., 2015). Here we reproduce Figure S1 of that paper where we demonstrate that MitoTracker Green is a faithful marker for mitochondrial mass when compared with the amount of mitochondrial DNA, CMXRos and other mitochondrial proteins. We pointed out to this reference when introducing MG as a marker, at the beginning of the section 'Mitochondrial content discriminates cell fate'

Fig S1

Figure S1. CmxRos is a good reporter of mitochondrial mass.

a In order to study the suitability of fluorescent probes for mitochondrial content quantification, we first sorted cells according to MitoTracker Green (MG) uptake in two subpopulations: High (H) and Low (L), inset in Figure S1a. Then, we quantified the relative mtDNA levels by qPCR as described in Supplementary Methods in both subpopulations. The ratios of MG staining and mtDNA levels between H and L populations show a perfect matching (Figure S1a). Error bars are the standard deviations of three biological replicates. **b** To check whether CmxRos is a faithful reporter for mitochondrial content in individual cells, we first co-stained cells with CmxRos and MG, showing a high correlation ($r^2=0.92$, Pearson correlation). In order to further validate the use of CmxRos we co-stained cells with CmxRos and several mitochondrial markers. **c-e** Three mitochondrial matrix proteins: Aconitase (ACO2), SOD2 and Pyruvate Dehydrogenase (PDP1) also correlate well with CmxRos: $r^2=0.88$ (Aconitase), $r^2=0.82$ (SOD2), $r^2=0.75$ (PDP1). **f** Complex I, which is located in the inner membrane, closely matches CmxRos levels ($r^2=0.88$). The antibodies used were ACO2 (AB110321), Complex-1 (AB109798), SOD2(AB110300) (Abcam) and PDH (2784) (Cell Signaling).

Reviewer #3 (Remarks to the Author):

In their manuscript, Diaz-Colunga and Márquez-Jurado and colleagues tackle the fundamentally important and clinically relevant problem of non-genetic heterogeneity in cellular responses to apoptotic stimuli. They show experimentally that in HeLa cells, the mitochondrial mass in a cell correlates with both its the TRAIL-induced cell fate as well as its abundance of proteins involved in the apoptotic response to TRAIL. Computationally, they show that recapitulating the distribution of mitochondrial mass and its correlation with the expression of regulators of extrinsic apoptosis is sufficient to recapitulate several features of the cellular response to TRAIL and its heterogeneity. Finally, the authors show that similar correlations between mitochondrial mass and the abundance of apoptotic regulator also exist in cells from colon cancer samples. Overall, this is a well written manuscript that addresses the important problem of fractional killing in TRAIL-treated cancer cells using well thought-out analyses based on solid data. The authors should address a few concerns to further strengthen their manuscript.

We thank the reviewer for her/his positive comments on our manuscript and for the thorough revision of the paper, that has helped us to improve the former version. Below follows a detailed answer to the comments raised by the reviewer.

Major issues

1. With regards to the simulations of the EARM model, there is one central assumption by this work that is glossed over in the manuscript: that the mitochondrial mass is equivalent to the concept of 'Pores'. Because of the central role of 'pores' in the initiation of the MOMP pathway, the authors should more explicitly analyse and discuss how much this assumption is central to their computational findings.

We apologize for any confusing statement or discussion in the previous version of our manuscript that may have given the impression that the identification of mitochondrial mass with the variable 'Pore' is a central assumption in the model.

In the EARM model, the variable 'Pore' represents the number of potential sites in the mitochondrial membrane where a pore could be formed. When a Bax tetramer binds to the mitochondrial membrane, the variable 'Pore' turns into an activated variable denoted as 'Pore' (reaction #19 in Supplementary Table 2) that actually represents a mitochondrial pore able to release Cytochrome C and Smac to the cytosol (reactions #20 and #21 in Supplementary Table 2). The limiting factor to the process is the availability of Bax, since the 'Pore' variable is in excess (in the model, and taking for the average value of this variable the one employed in previous applications of EARM, below 1% of all the potential sites are occupied).*

While it is true that in our simulations we sample the variable 'Pore' directly from the distribution of CMXRos levels (a reporter of mitochondrial mass) in a clonal population of cells, Figure 6b shows that the key assumption in the model is not the heterogeneity in the 'Pore' variable per se, but the correlation of mitochondria with the pro and anti-apoptotic proteins in the pathway. These correlations are able to explain by themselves the observed experimental dependence of apoptotic fate and death times on initial mitochondrial mass.

We have clarified this issue in the explanation of the Mathematical Model in the Supplementary Information, and revised the main text to avoid confusion.

2. With regards to the correlation of measured mitochondrial mass with cell fate and time to death (fig. 2), one alternative explanation for the data is that there is a certain toxicity from MitoTracker Green (and/or exposure to light in the presence of the dye), and that cells with more dye are more likely to die, and die early. Are the distributions of cell death times and the fraction of apoptotic cells affected by the use of MitoTracker Green?

We have included new experiments quantifying the fraction of apoptotic cells in samples treated with and without MitoTracker Green (see attached figure). We observe small differences in the control (no TRAIL), which could be due to a slight phototoxicity induced by the dye, but it is a minor effect (from 3% without dye to 6.4% with MG). For TRAIL treated samples we observe no significant difference between the cells that incorporated MG and those without the dye. We thus think that MG is not interfering with TRAIL to induce apoptosis and it is unlikely that mediates the observed bias in

mitochondrial content in apoptotic/survivor cells. We have included these experiments as a new Supplementary Fig. 4 in the modified Supplementary Information.

Other issues

3. p.6 – with regards to the validation of the long half-life of TRAIL in the culture supernatant, the authors cite ‘data not shown’, however, I believe the policy of Nature journals is to request that all supporting data be presented in main or supplementary figures.

We have included these data as a new Supplementary Fig. 1, and modified the main text accordingly.

4. p.7 – with regards to the discussion of the influence of the cell cycle on cell death, the analysis in supp. fig. 2c is well done, but the authors should clearly state how the first analysis in sup. fig. 2a is flawed and thus requires the simulation test done in supp. fig. 2c: in the way the data is collected, there must be a bias as observed in part 2a, as a cell certainly cannot have a long time to death if it has a short time to division.

We have modified the discussion of the possible influence of cell cycle on apoptosis in p. 7, and the order of Supplementary Fig. 2 (Supplementary Fig. 3 in the modified version) to better clarify this issue. With regard to the analysis shown in the modified figure, we stated that ‘These biases may ... be simply a consequence of the fact that cells with fast commitment to death after TRAIL addition do not have time to divide before dying’.

5. p.11 – with regards to evaluating time of cell death in simulation results, I assume that the time at which Smac protein reach 90% of ‘saturated’ cytosolic levels is evaluated only for cells which are deemed apoptotic?

This is right. First we assign a cell fate (survivor or apoptotic) based on the rate of Casp8 activation, and for those cells deemed as apoptotic we evaluate the apoptosis time from Smac dynamics. Our simulations show that cells classified as ‘apoptotic’ release around 60-100% of the Smac initially confined to mitochondria, while ‘survivor’ cells release between 0-10%.

Are similar ‘saturated’ cytosolic levels reached in all cells called apoptotic? If not, how is time of death impacted by this particular definition?

The amount of initial Smac, and thus the saturated levels of cytosolic Smac, is variable from cell to cell since we sample initial levels of Smac based on the experimental distributions constrained by mitochondrial mass. For cells classified as apoptotic, saturated cytosolic Smac levels roughly range from $5 \cdot 10^4$ to $2 \cdot 10^5$ molecules (see attached figure, where we show, in different colours, cytosolic Smac dynamics of several ‘apoptotic’ cells with different initial Smac levels). We notice that Smac release is a relatively fast process and death times are not very influenced by the particular choice of the 90% of saturated levels, but what it is really variable from cell to cell is the time elapsed to start release due to pore formation. This time lapse depends on the particular levels of all pro and anti-apoptotic proteins in the pathway.

6. p.12-3 – it should be made clearer that when performing the discrimination analysis for simulations accounting for co-variation with mitochondrial mass, Bid and Bax abundance, which have some discrimination power, are *not* independent from the abundance of other proteins in the system, while when sampling is done

independently, they are. Therefore, the discrimination power of Bid and Bax is likely linked to the fact that they can 'predict' the abundance of several proteins in the system.

We agree with the referee that when sampling is constrained by mitochondria-protein co-variation, protein abundances are interdependent. In section 'Mitochondria-protein correlations constrain protein-protein correlations' of the modified Supplementary Information, we estimate protein-protein correlations due to co-variation of each protein with mitochondrial mass. Even in the absence of other common explanatory variables or interactions, the expected protein-protein correlations vary between 0.5 and 0.8 (Pearson correlation). In this sense, we agree that the discrimination power of proteins such as Bid and Bax is linked to the abundance of other proteins, as the reviewer points out, but we believe that the underlying predictive or independent variable is mitochondrial mass. This is the fact that we wanted to stress with the simulations in Figure 6b.

7. p. 13 – it should be made clear that the negative sensitivity of discrimination performance for Bcl-2 and Mcl-1 does not necessarily reflect that they are not important to the cell fate, but rather that a higher correlation with mitochondrial protein mass effectively couples their abundance to that of pro-apoptotic proteins, and thus reduces discriminating power.

This is indeed a plausible explanation for the negative sensitivity of discrimination to mitochondrial correlation observed in anti-apoptotic proteins. We have included the reviewer's observation in the discussion of Figure 6c, pp. 13-14.

In the Discussion section we also connected this observation (the opposed sensitivity of pro and anti-apoptotic proteins) to the fact that pro-apoptotic proteins like Bax and Bid are markers for apoptotic 'priming' or susceptibility, while the anti-apoptotic proteins Bcl-2 and Mcl-1 may act as 'buffers' of apoptotic activity (Sarosiek, K. A., Ni Chonghaile, T. & Letai, A. Trends Cell Biol 23, 612-619 (2013)), which indicates that their levels should be de-regulated from those of pro-apoptotic proteins, in line with the referee's suggestion.

8. p.17 (lines 403-412) – the various connections between mitochondrial mass, mitochondrial function and mitochondrial gene expression and certain cell fates are not necessarily support for the correlation between mitochondrial mass and propensity for apoptosis, but rather interesting and important contexts in which this connection certainly merits investigation.

Agreed. We have changed the first sentence of the paragraph accordingly.

9. p.17 – while the data presented in the manuscript does suggest that mitochondrial content could be a good biomarker for sensitivity to TRAIL, but as TRAIL itself is not at this time a clinically relevant cell death stimulus, the connection should be

demonstrated for other pro-apoptotic stimuli or clinically relevant anti-cancer drugs before an extensive analysis is done in cancer samples, as suggested.

As also suggested by another reviewer, we have extended the basic analysis of apoptotic cell fate and mitochondrial mass to other apoptotic inducers of death in HeLa cells, one from the extrinsic program (TNF) and two for the intrinsic one (DRB and Cycloheximide), shown as Supplementary Fig. 5. We also remarked in the final sentence of the Discussion that a more extensive analysis in different types of cancers and with different chemotherapeutic drugs must be done to be sure of the clinical value of mitochondria as biomarker.

Reviewers' Comments:

Reviewer #1:

Remarks to the Author:

Also if in the clone of HeLa cells of the authors the long form is the only present, at least the introduction section should be implemented by adding a brief overview of Mcl-1 anti/pro-apoptotic functions related to its subcellular localization and specific forms (PMIDs: 26538029, 19683529, 24260268, 20540941)

Reviewer #2:

Remarks to the Author:

The ms. by Márquez-Jurado is much improved, however I have remaining issues regarding the BAX overexpression expt. both technically but more importantly in the context of the proposed model.

My interpretation of the new data is that it shows the correlation between mitochondrial content and apoptotic sensitivity persists even in the presence of overexpressed BAX. The authors state that this is due to mitochondria being global regulators of gene transcription, however the key proteins that would protect against BAX activity would be anti-apoptotic BCL-2 proteins, it's challenging to conceive how these would be at sufficient levels to protect against overexpressed BAX mediated killing. Rather than mitochondrial content correlating with apoptotic sensitivity due to alterations in pro-/apoptotic BCL-2 levels, it seems more likely that alterations in upstream effectors (e.g. caspase-8/FADD/receptor levels) are more relevant for dictating cell death sensitivity - this point should be clarified, given that the central focus is on mitochondrial content regulating cell death sensitivity via pro-/apoptotic BCL-2 levels. Presumably, if it is dictated by BCL-2 levels, intrinsic apoptosis triggers would also display increased sensitivity in mitochondria high cells.

From a technical perspective, many labs have shown that transient overexpression of BAX leads to its activation (likely transfection stress leads to its activation), it's unclear whether this was observed in the described expt. but stable expression would have been better. More importantly, it's difficult to see how the % cell transfection rate was obtained given that the Bax appears untagged, this is important since obviously high Bax expression in a low % of cells (due to assay/model sensitivity issues) could lead to the erroneous conclusion that mitochondrial content and apoptotic sensitivity persists even in the presence of overexpressed BAX

Reviewer #3:

Remarks to the Author:

With their revision to their manuscript, Diaz-Colunga and Márquez-Jurado and colleagues have addressed all of my initial concerns, both minor and major, and seem to have satisfactorily addressed the concerns of the other reviewers. The additional data and analyses are well done, adding support and additional focus to the original conclusions, and the text changes have clarified the few points that were more difficult to understand in the initial submission. Overall, these changes have further improved the paper, and I recommend it for publication.

Reviewer #2 (Remarks to the Author):

The ms. by Márquez-Jurado is much improved, however I have remaining issues regarding the BAX overexpression expt. both technically but more importantly in the context of the proposed model.

My interpretation of the new data is that it shows the correlation between mitochondrial content and apoptotic sensitivity persists even in the presence of overexpressed BAX. The authors state that this is due to mitochondria being global regulators of gene transcription, however the key proteins that would protect against BAX activity would be anti-apoptotic BCL-2 proteins, it's challenging to conceive how these would be at sufficient levels to protect against overexpressed BAX mediated killing.

This is not the case. As we showed in the attached Figure, cells overexpressing Bax die around a 30% more than control cells. Thus, despite inactivated Bax levels being in excess, overexpression of Bax changes the balance between pro and antiapoptotic proteins favouring cell death.

c

Rather than mitochondrial content correlating with apoptotic sensitivity due alterations in pro-/apoptotic BCL-2 levels, it seems more likely that alterations in upstream effectors (e.g. caspase-8/FADD/receptor levels) are more relevant for dictating cell death sensitivity - this point should be clarified, given that the central focus is on mitochondrial content regulating cell death sensitivity via pro-/apoptotic BCL-2 levels. Presumably, if it is dictated by BCL-2 levels, intrinsic apoptosis triggers would also display increased sensitivity in mito high cells.

We apologize if our discussion about the sensitivity of cell death to correlation with mitochondria was not clear enough. In Figure 6b of main text we show that the discriminatory power of most of the apoptotic proteins (except that of Caspase-8) is lost if we do not take into account the correlations with mitochondria. Therefore, as we emphasize in the main text (second paragraph, section 'Mitochondrial heterogeneity,

key in variable TRAIL response), there are no particular key apoptotic proteins determining cell death, but the global interplay between all their levels modulated by mitochondria is the relevant determinant of cell fate.

Perhaps it is also confusing the use of the word 'sensitivity' to study the influence of co-variation of mitochondria with the levels of specific apoptotic proteins (Figure 6c of main text). Here we define sensitivity as the change in discriminatory capacity if the correlation of mitochondria with one particular protein at a time is varied. Here we find that the pairs of pro and anti-apoptotic proteins Bax/Bcl-2 and Bid/Mcl-1 are especially sensitive to this co-variation, which is different than sensitivity to cell death.

From a technical perspective, many labs have shown that transient overexpression of BAX leads to its activation (likely transfection stress leads to its activation), its unclear whether this was observed in the described expt. but stable expression would have been better. More importantly, its difficult see how the % cell transfection rate was obtained given that the Bax appears untagged, this is important since obviously high Bax expression in a low % of cells (due to assay/model sensitivity issues) could lead to the erroneous conclusion that mitochondrial content and apoptotic sensitivity persists even in the presence of overexpressed BAX.

Our first approach to Bax overexpression was done by transfection of the plasmid hBaxC3-EGFP (Addgene #19741) in which Bax is tagged with EGFP. This Construct allowed us to estimate the transfection efficiency, which reached 80-85% of total cells. However, this plasmid presents two technical problems. On one hand, the tag is green hampering the use of MitoTracker Green to simultaneously measure the single-cell level of mitochondrial mass and Bax. On the other hand, we were concerned about the possible interference of the EGFP tag with the function of Bax. For that reason, we decided to remove it. Nevertheless, we performed the TRAIL-induced apoptosis experiments obtaining the same fractions of death cells for both tag and untagged Bax.

As the referee correctly points, the transfection experiment with Bax has a series of detrimental effects. We observed that those cells expressing Bax in higher amount died in the first 48h, perhaps due to the self-activation of the intrinsic apoptotic program.

For those reasons, we followed the following protocol: First we transfected cells with the appropriate amount of DNA and transfection reagent to achieve the higher number of transfected cells but trying to avoid a massive overexpression that kills instantaneously the cells. Cells were transfected during 48 hours. During this time, a high number of cells died due to toxicity and self-activation, remaining in the plate those cells with exogenous levels similar to physiological ranges (this was evaluated with the EGFP-Bax). Then, the remaining cells were stained with MitoTracker Green and treated with TRAIL at the appropriate dilution. As we mentioned before, for both constructs, tagged and untagged Bax, we obtained similar fraction of death cells, being higher in cells overexpressing Bax than control cells.

We measured the level of Bax overexpression by Western Blot, which was around 3-4 times higher in transfected cells. Moreover, we analyzed by immunofluorescence the correlation between Bax (both exogenous plus endogenous) and mitochondrial

mass. We observed that in overexpressed cells the correlation was lower than in control cells, but still high.

This is not surprising, because the expression of exogenous Bax (under the control of a cytomegalovirus promoter) depends on the global regulation of gene expression. Therefore, mitochondria, through its role in energy production also control the exogenous gene expression, keeping an accurate correlation with Bax.

We ruled out the possibility to develop a stable Bax overexpressing cell line, because we reasoned that during the process of selection we will eliminate the cells with higher levels of expression, ending up with cells with levels of Bax very similar to the non-transfected cells.

Therefore, as we say above, even in Bax overexpressing cells, the correlation with mitochondria is still high and consequently, its sensitivity to co-variation with mitochondria amount persisted.